# Discounting and Drug Seeking in Biological Hierarchical Reinforcement Learning

**Vardhan Palod**[1]
School of Computing and Augmented Intelligence
Arizona State University, USA
vpalod@asu.edu

**Pranav Mahajan**
Nuffield Department of Clinical Neurosciences
University of Oxford, U.K.
pranav.mahajan@ndcn.ox.ac.uk

**Veeky Baths**
Cognitive Neuroscience Lab
APPCAIR
Department of Biological Sciences
BITS Pilani K. K. Birla Goa Campus, India
veeky@goa.bits-pilani.ac.in

**Boris Gutkin**
Laboratoire de Neurosciences Cognitives & Computationnelles
École Normale Supérieure Paris, France
boris.gutkin@ens.fr

## Abstract

Despite a strong desire to quit, individuals with long-term substance use disorder (SUD) often struggle to resist drug use, even when aware of its harmful consequences. This disconnect between explicit knowledge and compulsive behavior reflects a fundamental cognitive-behavioral conflict in addiction. Neurobiologically, differential cue-induced activity within striatal subregions, along with dopamine-mediated connectivity from the ventral to the dorsal striatum, is a key factor in driving compulsive drug-seeking. However, the functional mechanism linking these neuropharmacological findings to the cognitive-behavioral conflict remains unclear.

Another key aspect of addiction is temporal discounting, with studies showing that individuals with drug dependence exhibit steeper discount rates than non-users. Assuming the ventral-dorsal striatal organization reflects a gradient from cognitive to motor-action representations, addiction can be modeled within a hierarchical reinforcement learning (HRL) framework. However, incorporating discounting into the biological HRL framework is challenging, and remains an open problem.

In this work, we build upon an algorithmic model that captures how the action choices that the agent makes when reinforced with drug rewards become impervious to the presence of negative consequences that often follow those choices. We address the challenge of incorporating discounting into the HRL framework by ensuring that the values of natural rewards converge across all hierarchical levels in the HRL framework. In contrast to natural reward values, we show that the pharmacological effects of drugs on the dopamine system cause divergence in drug reward values.

Our results demonstrate that high discounting amplifies drug-seeking behavior across all levels of the hierarchy, suggesting that faster discounting is associated with increased addiction severity and impulsivity. We show how these results align with the evidence supporting temporal discounting as a behavioral marker. Additionally, our model offers testable predictions and establishes a framework that conceptualizes addiction as a disorder of hierarchical decision-making processes.

**Keywords:** Hierarchical Reinforcement Learning; Addiction; Impulsivity; Temporal discounting; Dopamine.

## Introduction

Drug addiction is characterized by the persistent and compulsive pursuit of drug rewards, often in the face of severe adverse consequences. A signature of such pathological behavior becomes evident in controlled experiments where addicts exhibit a characteristic "self-described mistake": an inconsistency between the potent behavioral response toward drug-associated choices and the relatively low subjective value that the addict reports for the drug (Goldberg, 1991; Stacy & Wiers, 2010; Goldstein et al., 2010). Several studies have proposed that prolonged exposure to drugs when coupled with the loss of inhibitory cognitive control over behavior could be responsible for the transition from casual to compulsive drug-seeking behavior (Everitt & Robbins, 2005; Kalivas & Volkow, 2005; Belin et al., 2009; Keramati & Gutkin, 2013). The loss of cognitive control and self-acknowledged mistakes in addiction remain inadequately explained by formal computational models (Redish, 2004; Takahashi et al., 2008; Dezfouli et al., 2009; Dayan, 2009; Piray et al., 2010; Garrett et al., 2023) and animal models (Ahmed & Koob, 1998; Gardner, 2020; Hogarth, 2020). Amongst several computational approaches, one dominant approach utilises the model-free reinforcement learning framework to explain addiction by interpreting it as a maladaptive state of the habit learning system (Redish, 2004; Takahashi et al., 2008; Dezfouli et al., 2009; Dayan, 2009; Piray et al., 2010). These models propose that drugs, through their pharmacological effects on dopamine signaling, believed to convey a teaching signal, cause excessive reinforcement of drug-seeking actions, which in turn lead to compulsive drug-seeking habits. Although this simplified perspective has addressed certain aspects of addiction, growing evidence suggests that multiple learning systems contribute to the pathology.

In this paper, we aim to develop a decision-making model that explains addictive behavior using the Hierarchical Reinforcement Learning (HRL) framework whilst incorporating the effects of temporal discounting. HRL is an extension of tra-

---

[1]Majority of this work was done during his undergraduate studies at BITS Pilani University, K. K. Birla Goa Campus, India.

ditional reinforcement learning (RL) that incorporates a hierarchical structure into the decision-making process. This reflects how humans intuitively use abstractions, for instance, the goal of getting coffee may begin with identifying the nearest coffee shop. This is followed by a sequence of mid-level actions, such as walking to the door, navigating streets, entering the café. At an even finer level, each of these actions is further broken down into primitive motor actions. Botvinick et al. (2009) posit that HRL provides a compelling computational framework for explaining the neural underpinnings of complex, hierarchically structured behaviors. Eckstein & Collins (2020) demonstrate that hierarchical abstraction significantly improves modeling of human learning, implying that the brain likely leverages HRL-like organization (a "biologically inspired" approach) to achieve flexible and efficient learning in complex environments.

Our hierarchical reinforcement learning framework assumes that an abstract cognitive plan is broken down into a sequence of lower-level actions, ultimately culminating in concrete lowest-level responses at the base of the hierarchy (Botvinick, 2008; Botvinick et al., 2009). Neurobiologically, the decision-making hierarchy, from cognitive to motor levels, is organized along the rostro-caudal axis of the cortico-basal ganglia (BG) circuit (Koechlin et al., 2003; Badre et al., 2009; Badre & D'esposito, 2009). This circuit comprises multiple parallel, closed loops connecting the frontal cortex with the basal ganglia (Alexander et al., 1986, 1991). While the anterior loops are responsible for representing more abstract actions, the caudal loops that include the sensory-motor cortex and the dorsolateral striatum are associated with encoding low-level habits (Koechlin et al., 2003; Badre et al., 2009; Badre & D'esposito, 2009). Midbrain dopamine (DA) neurons signal reward prediction errors to the striatum, reinforcing stimulus-response associations (Schultz et al., 1997). Through spiraling connections, DA projections link ventral and dorsal striatal regions, enabling feed-forward coupling across cortico-basal ganglia loops (Haber et al., 2000; Haber, 2003; Belin & Everitt, 2008). These DA spirals allow the abstract cognitive levels of valuation to guide the learning in the more detailed action-valuation processes (Haruno & Kawato, 2006).

Keramati & Gutkin (2013) presents a hierarchical reinforcement learning (HRL) model where higher levels represent abstract, cognitive options and lower levels represent more primitive actions. Learning occurs faster at higher levels due to fewer abstract states, and prediction errors propagate through the hierarchy, influenced by drug-induced biases. Their model focuses on value learning without addressing action selection. Mahajan et al. (2023) extend this work to an HRL algorithm which captures the imbalanced decision hierarchy emerging in an agent with substance use disorder. However, neither the Keramati & Gutkin (2013) model nor the Mahajan et al. (2023) model includes temporal discounting of future rewards. Both focus on hierarchical value learning and the propagation of prediction errors, emphasizing the impact of drugs on compulsive behavior without incorporating temporal discounting of

rewards and punishments that follow. In our framework, compulsivity refers to the emergence of rigid, habitual stimulus-response behaviors at lower levels of the hierarchy, consistent with prior HRL models (Keramati & Gutkin, 2013).

Temporal discounting, at a behavioral level, refers to the decrease in the perceived value of a reward with the time delay associated with its reception (Ainslie, 1975; Rachlin & Green, 1972). Numerous studies have shown that people suffering from substance use disorders (SUDs) overvalue immediate, drug-associated rewards and undervalue long-term natural rewards (Bickel et al., 2007; Schultz, 2011). Impulsivity can be defined as the degree to which an individual disproportionately favors short-term rewards over long-term outcomes. Bickel et al. (2014) proposed that the degree of discounting can be used as a behavioral marker to indicate the addiction severity in people with SUDs and as a predictive measure indicating susceptibility to developing SUDs.

Motivated by these studies, we present a novel method to incorporate discounting in HRL modeling framework. We begin by demonstrating the challenges involved in incorporating discounting into biological HRL. Then, we introduce our approach and illustrate how it implements normative discounting. We then use this model to analyze shifts in an agent's preferences in response to varying discounting factors. Finally, we show how the HRL framework explains the cognitive-behavioral conflict in addiction and aligns with the findings of Bickel et al. (2014).

## Results

### Challenges with incorporating discounting in biological hierarchical reinforcement learning

Hierarchical reinforcement learning (HRL) is formalized using semi-Markov Decision Processes (semi-MDPs), which support temporal and state abstraction via the options framework Precup (2000). In this framework, lower levels correspond to primitive actions (e.g., motor movements), while higher levels represent temporally extended options composed of those actions. For instance, in the task of "getting coffee," a primitive action might involve moving a hand, while an option may encapsulate walking to the kitchen. At each level, the Q-value of a state-option pair captures the expected cumulative reward upon initiating and following through with that option.

In the model proposed by Keramati & Gutkin (2013), higher levels are viewed as more "cognitive" and lower levels as more "habitual," though this distinction does not imply goal-directed behavior. Drug-induced effects are modeled by adding a non-negative bias $d$ to the temporal-difference (TD) error, capturing elevated striatal dopamine responses Redish (2004); Dezfouli et al. (2009); Piray et al. (2010); Dayan (2009); Di Chiara & Imperato (1988). Unlike the non-converging TD errors in Redish (2004) due to the max operator, the HRL framework allows error convergence while modeling persistent drug effects. A full discussion of how the HRL model addresses prior limitations is provided in Mahajan et al. (2023).

Following Haruno & Kawato (2006), the TD error at abstrac-

tion level $n$ is computed using value information from level $n+1$:

$$\delta_t^n = [r_t^n + V^{n+1}(s_{t+1}^{n+1}) - Q^n(s_t^n, a_t^n)] + d$$
$$= [r_t^n + Q^{n+1}(s_t^{n+1}, a_t^{n+1}) - r_t^{n+1} - Q^n(s_t^n, a_t^n)] + d \quad (1)$$

Here, $a_t^{n+1}$ denotes the abstract option, and $a_t^n$ is the corresponding primitive action. Likewise, $r_t^{n+1}$ incorporates the reward from $r_t^n$. If the action advances the agent toward a natural reward, $d = 0$; for drug-directed behavior, a positive bias $d = +D$ is applied. Further implementation details are provided in the Methods section.

To motivate why incorporating discounting into HRL discounting is challenging, consider that time tends to move more slowly at higher levels than at lower levels. One abstract option in a higher-level decision corresponds to multiple primitive steps at a lower level. This disparity makes it difficult to apply discounting consistently across different levels of the hierarchy, as the time span of each decision expands with abstraction. An example is illustrated in Figure 1b, the option $b_{t+1}^{n+1}$ corresponds to the combination of primitive steps $a_{t+1}^n$ and $a_{t+2}^n$.

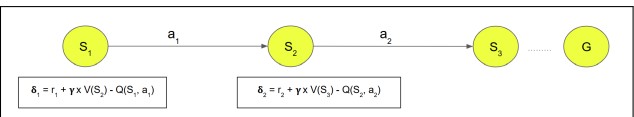

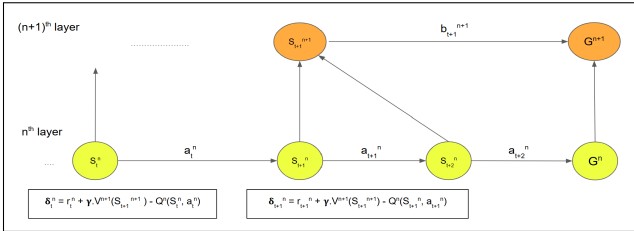

(a) Q-value update rule in standard (flat) RL

(b) Q-value update rule in HRL with state abstraction (naive discounting with $\gamma$ as the discounting factor at the $n^{th}$ level)

Figure 1: Comparison of Q-value updates in standard RL versus HRL with state abstraction. (a) In standard RL, the Q-value of a state updates based on its temporal relationship with the subsequent state. (b) In HRL with state abstraction, Q-values update according to Equation 1, relying on abstract states at a higher level rather than consecutive states at the same level. This abstraction removes the temporal component, making consecutive states have identical Q-values, irrespective of their relative proximity to the goal. This highlights the challenge of incorporating discounting in HRL.

Additionally, the difficulty arises from the way the states in the $n^{th}$ level learn the action values from the abstract states in the $(n+1)^{th}$ level and not from the subsequent states in the $n^{th}$ level (RPE is calculated according to equation 1).

The temporal component that exists when states learn from

their immediate subsequent states is removed. In fact, two consecutive states at a lower level mapped to the same abstract state at a higher level will have the same action values even though one of them is closer to the goal state. In Figure 1b, $Q(S_t^n, a_t^n)$ and $Q(S_{t+1}^n, a_{t+1}^n)$ will be equal as their respective successors i.e $S_{t+1}^n$ and $S_{t+2}^n$ and are mapped to the same abstract state $S_{t+1}^{n+1}$ even though $S_{t+1}^n$ is closer to the goal state $G^n$ than $S_t^n$.

A naive approach of adding a discount factor in equation 1 modifies the equation as -

$$\delta_t^n = r_t^n + \gamma(Q^{n+1}(s_t^{n+1}, a_t^{n+1}) - r_t^{n+1}) - Q^n(s_t^n, a_t^n) + d \quad (2)$$

where $\gamma$ is the discounting factor.

To illustrate that an approach with a single discounting factor for all the hierarchy levels is insufficient, we simulate an agent based on this naive implementation in a two-choice task environment, that has to choose between a food reward and a drug reward (Figure 2a). Note that drug rewards, unlike the (natural) food rewards, are followed by an unavoidable punishment and also incorporate the hijacking effect in TD errors as mentioned before.

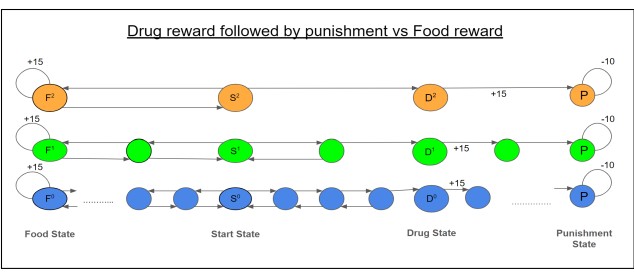

(a)

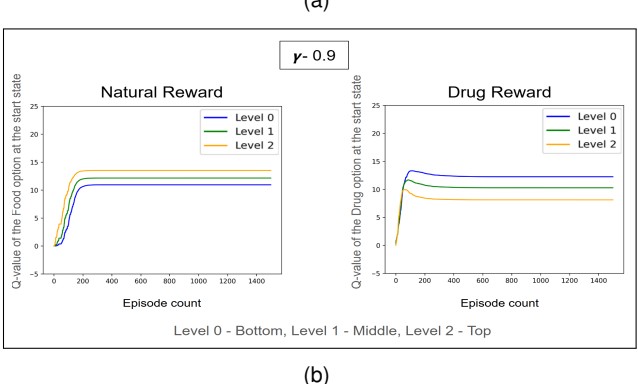

(b)

Figure 2: (a) The two-choice task environment where the agent chooses between a Food reward of $+15$ vs. a Drug reward of $+15$ followed by a punishment of $-10$ with an additional drug-associated $D = 3$ effect on RPE. (b) Q-values of actions leading to Food and Drug rewards at the start state of the environment. The experiments were conducted in the two-choice MDP shown in (a).

By examining the Q-values of the options at different levels, to select food or drug rewards, starting from the starting state $S$, we find that discounting according to equation 2 leads to

the deviation in the Q-values even for actions leading to natural rewards (Figure 2b, left). The model valuation predicts that the Q-values of actions associated with natural rewards decrease as we move down the hierarchy. Firstly, this result is not normative: the valuation of the same outcome should not differ merely due to changes in the level of abstraction. Secondly, it contradicts the hypothesis proposed by Keramati & Gutkin (2013); Mahajan et al. (2023), which suggests that hierarchical valuation inconsistencies arise specifically due to the pharmacological effects of drugs, not in the case of natural rewards. The divergences in Q-values of drug-seeking options in the model (Figure 2b, right), include opposing effects of (intended) drug-hijacking effect inflating the values for lower levels and (unintended) improper discounting shrinking the values for lower levels.

Motivated by these reasons, we developed a novel model that integrates discounting across hierarchical levels, addressing the complexities of spatial and temporal abstraction in the HRL environment. We first test if this model normatively captures discounting solving the problem of convergence in natural (food) rewards. Then we show the predictions of the model with drug rewards.

**Normative discounting in the biological HRL model**

We extend the model proposed by Mahajan et al. (2023) by incorporating discounting across all levels of the hierarchical decision structure. Each level is assigned an adjusted discount factor that is calculated according to the discount factor at the topmost level, which we refer to as the effective discount factor $\gamma$, and the number of intermediate steps introduced at each level $\vartheta_n$. The adjusted discount factor at level $n$ is denoted by $\zeta(n)$ as follows.

Let the $L$ levels in the hierarchy be zero-indexed as follows, $n = 0, 1, .., L-1$, where level $n = 0$ is the bottom-most level and the level $L-1$ is the top-most level. We further define $\vartheta_n$ as the number of consecutive options in level $n$ required to construct the corresponding option in level above $n+1$. For the topmost level, $\vartheta_{L-1}$ is set to 1. For example, in our Figure 2a, we have 3 levels $n = 0, 1, 2$ and each option decomposes into two smaller options at the lower level, therefore $\vartheta_0 = 2, \vartheta_1 = 2, \vartheta_2 = 1$. Further, we let the effective discount factor be $\gamma$, which is equivalent to the discount factor used in the topmost level. This allows us to propose an equation for the adjusted discount factors $\zeta(n)$ as a function of level $n$ and the effective discount factor $\gamma$, ensuring that values at different levels do not diverge, but rather converge according to the effective discount factor $\gamma$.

$$\zeta(n) = \gamma^{\left(\prod_{i=n}^{L-1} \frac{1}{\vartheta_i}\right)} \quad (3)$$

Further, if the $\vartheta_n$ are the same across all levels (except the topmost level which is 1), then we can set it to simply $\vartheta$ i.e. at every level an option decomposes into $\vartheta$ number of smaller options at the level below. This further simplifies the equation for adjusted discount factors as follows,

$$\zeta(n) = \gamma^{\left(\frac{1}{\vartheta_i}\right)^{L-1-n}} \quad (4)$$

Therefore in the case presented in Figure 2a, if we use the effective discount factor as $\gamma = 0.9$ and $\vartheta = 2$, then the adjusted discount factors for the three levels are $\zeta(n=0) = (0.9)^{\frac{1}{4}}, \zeta(n=1) = (0.9)^{\frac{1}{2}}, \zeta(n=2) = 0.9$.

This adjustment to the discount factors at each level ensures that the model remains consistent with the effective discount rate $\gamma$. If we simply had hierarchical levels, with varying granularity in options, but did not incorporate biologically plausible spiraling connections then we could have directly used adjusted discount factors $\zeta(n)$ at respective levels. But in the biological HRL framework that we work with, the TD-errors require using the Q-values from a level above, representing the spiraling connections across the striatum. Therefore, to use these adjusted discount factors to compute TD-errors, we first need to "undiscount" the values from the level above $(n+1)$ before discounting them appropriately at the current level ($n$), as follows:

$$\delta_t^n = r_t^n + (\zeta(n)^{\nu(s_t^n, a_t^n)}) \cdot \frac{Q^{n+1}(s_t^{n+1}, a_t^{n+1}) - r_t^{n+1}}{(\zeta(n+1))^{\nu(s_t^{n+1}, a_t^{n+1})}}$$
$$- Q^n(s_t^n, a_t^n) + d \quad (5)$$

where $\nu(s_t^n, a_t^n)$ indicates how many steps the state $s_t^n$ is from reaching the goal state associated with the action $a_t^n$ (explained further in Methods, using Figure 6), within that level $n$. At each level, the Q-values for state-action pairs, $Q(s_t, a_t)$, and the state-function values, $V(s_t)$, adjust to their discounted values based on the discount factor $\zeta(n)$ for that level.

This structure requires the system to account for the steps to future rewards at respective levels of hierarchy when calculating the reward prediction error (RPE). Such information-sharing may be biologically plausible as recent research indicates that midbrain dopamine neurons may encode information about the distribution of time between the cue presentation and the rewarding outcome (Sousa et al., 2023).

First, we test whether our model reproduces normative behavior and solves the problem of lack of convergence of Q-values for actions leading to natural rewards. We will discuss the drug reward scenario later. We test this in an environment where the agent has to choose between an immediate food reward vs. a delayed food reward of equal magnitude as shown in Figure 3a.

We find that the HRL model based on equation 5 performs discounting normatively, as demonstrated by the results in Figure 3b. The Q-values of actions leading to food rewards converge across all the levels in the hierarchy. The Q-value of the delayed reward is further discounted and converge to a lower value as compared to the value of immediate reward and thus, the agent is performing discounting normatively. We conclude that this method provides a foundation for further investigating the effects of discounting within the framework of HRL modeling in addiction.

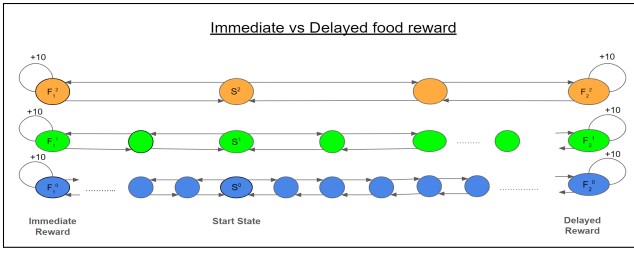

Immediate vs Delayed food reward

(a)

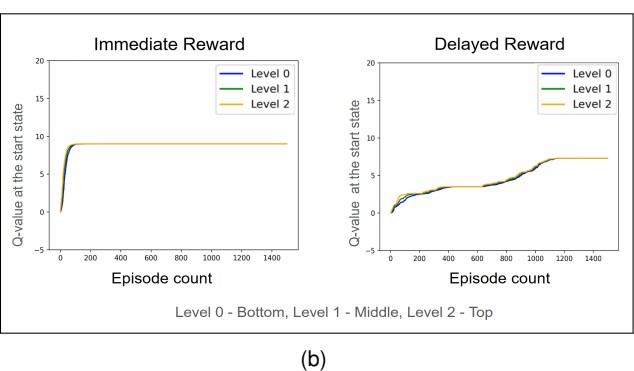

Level 0 - Bottom, Level 1 - Middle, Level 2 - Top

(b)

Figure 3: (a) The two-choice task environment where the agent chooses between an immediate vs. delayed $+10$ food reward. (b) Q-values of actions leading to immediate and delayed food rewards at different levels when $D = 0$ and $\gamma = 0.9$. Across all levels, the Q-values converge to 9 for the immediate reward and to 8.1 for the delayed reward.

## Model prediction: Over-discounting exacerbates drug-seeking behavior

Having implemented normative discounting in natural rewards, we now turn to model predictions in the case of drug rewards. Temporal discounting, where delayed rewards are perceived as less valuable, is significant in understanding substance use disorders (SUDs), as individuals with SUDs tend to overvalue immediate drug rewards and undervalue long-term natural rewards, with the degree of discounting serving as a marker of addiction severity (Bickel et al., 2007, 2014; Schultz, 2011).

In this experiment, we aim to examine how variations in the discounting rate influence the agent's drug-seeking behavior across different levels of the hierarchy in the HRL framework. We hypothesize that more discounting will lead to more drug-seeking behavior. To study the preference of the agent towards drug rewards over natural rewards, we trained our HRL agent in a two-choice environment where it has to choose between a food reward or a drug reward followed by a punishment (Figure 2a). We use hard allocation, clamping behavioral control to a single hierarchical level to assess agent behavior under full control of option selection at that level (other approaches to allocating control over levels are discussed by Mahajan et al. (2023)). The controlling level selects actions or options using Boltzmann exploration based on the Q-values of the corresponding state-option pairs. Each episode ter-

minates when the agent either reaches the reward states or exceeds the maximum step limit. (Please refer to Methods section for a detailed description of the environment, agent design and action selection). Across trials, we increased the discounting rate by adjusting the discount factor $\gamma$ from 1 to 0.8 and 0.6 to examine its effect on the agent's drug-seeking propensity.

We find that irrespective of the value of the discounting factor, drug-seeking increases as we move down the hierarchy. However, for $\gamma = 1$, the increase is minimal when comparing drug-seeking behavior between levels 2 and 1 (Figure :4a). The combined effect of discounting and the accumulation of the pharmacological $+D$ factor make the drug action the favourable choice of the agent when the agent's behavior is controlled by the lower levels of the hierarchy. Similar effects of increased drug-seeking as the behaviour control is transferred to lower levels were observed in the previous versions of the model (Keramati & Gutkin, 2013; Mahajan et al., 2023) which did not involve temporal discounting. Our model extends these previous models to additionally incorporate the effect of temporal discounting, by demonstrating that over-discounting further increases drug seeking even when it is followed by punishment. As seen in Figure 4a, increasing discounting in turn significantly increases drug-seeking at every level in the hierarchy.

## Selective delay of natural rewards vs. immediate drug rewards

In the real world, natural and drug rewards are often not equidistant, requiring different times to obtain. Often, natural rewards that offer greater benefit require long-term planning, whereas drug seeking actions are often short sighted which provide immediate reinforcement but are frequently followed by negative consequences. To simulate these conditions, we studied the agent's preference in a two-choice environment where a drug reward is available immediately and is followed by a punishment, whereas a more advantageous natural reward is available after a delay of 1 time-step.

Our findings (Figure 5b) show that in the scenario under consideration, at lower levels the agent always prefers the immediate drug reward over food reward irrespective of the $\zeta(n)$. This result is different than the one which we found in the previous environment where the food reward and drug reward were equidistant. Based on the agent's behavior, we can conclude that the agent inherently becomes more impulsive when the behavior is controlled by the lower levels, in this task. Even when the higher layers are controlling the behavior, we observe a similar gradient of increasing impulsivity as discounting increases as seen in the Figure 4b. Our model suggests that the cumulative effect of drug rewards is sufficient to drive the agent toward drug-seeking behavior when decision-making is governed by lower levels of the hierarchy. Additionally, our model predicts that increased discounting at higher levels enhances the agent's impulsivity, further reinforcing the preference for immediate drug rewards.

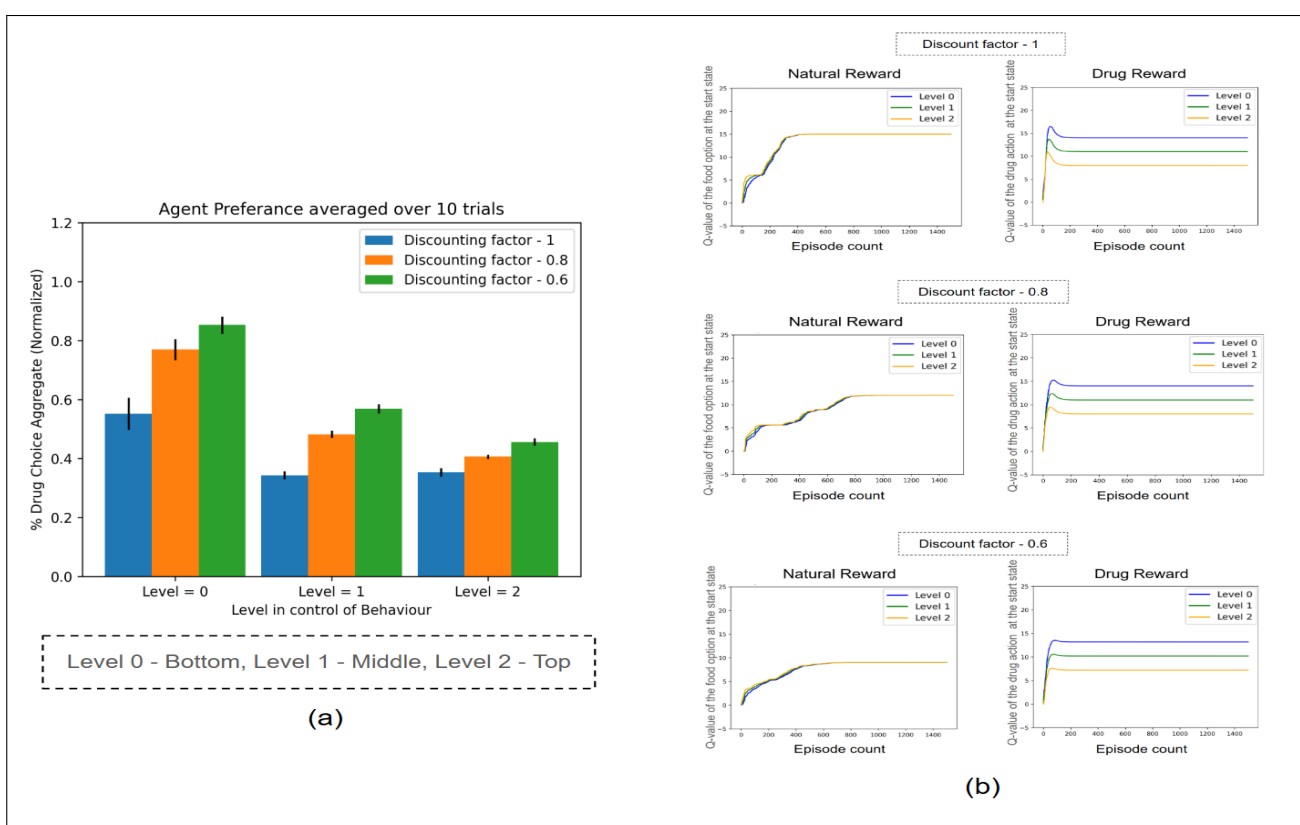

Figure 4: This figure shows the simulation results of the two-choice task (Figure 2a) where drug reward is followed by punishment. (a) The bar chart illustrates the comparison between % drug seeking by the agent when discounting is varied. In all cases, drug-seeking increases when the lower levels of the hierarchy are in control of behavior. When discounting is increased, drug-seeking significantly increases at all levels of the hierarchy. (b) Q value for food/drug-seeking option at the start state on the y-axis and are calculated through algorithmic simulations involving environment interactions, with $D = 0$ for Natural reward and $D = +3$ for drug reward, from equation 7, following the simulation protocols described in section 4.4. The error bars represent the variability across trials, which arises due to stochasticity in the action selection process of the agent which is based on Boltzmann exploration.

## Discussion

Previous HRL models (Keramati & Gutkin, 2013; Mahajan et al., 2023) provided a normative framework for understanding the distinct roles of ventral and dorsal striatal circuits in drug-seeking acquisition and habit execution, as well as the selective influence of feed-forward dopamine connectivity on drug versus natural reinforcers. However, these models lacked a critical component: Temporal discounting, which is essential to simulating real-world decision making. Incorporating discounting into HRL frameworks has long been a challenge. Harutyunyan et al. (2019); Hengst (2003) proposed methods for discounting within the Options framework and using state abstraction. However, no existing approaches in the literature address discounting within a biologically plausible HRL framework. In this work, we address this gap by introducing a novel approach to incorporate discounting into the biological HRL framework (Haruno & Kawato, 2006; Keramati & Gutkin, 2013; Mahajan et al., 2023).

Steep delay discounting is a common trait observed in various substance use and psychiatric disorders, including gam-

bling addiction, drug dependence, and internet gaming disorder Amlung et al. (2017, 2019); Wöhlke et al. (2021). Amlung et al. (2017) conducted a comprehensive meta-analysis and found a significant association between steeper delay discounting and greater substance use frequency/quantity as well as a slightly higher correlation with indices of SUD problem severity. They concluded that delay discounting is robustly associated with continuous measures of addiction severity, and this relationship holds across different types of addictive substances and behaviors. Bickel et al. (2014) proposed temporal discounting as a behavioral marker for addiction, highlighting its potential to: 1) identify individuals at risk of developing drug dependence, 2) measure the severity of addiction, and 3) connect to the biological and genetic mechanisms underlying addiction. Our model aligns with these predictions, demonstrating that agents with higher discounting rates exhibit an increased propensity for drug-seeking behavior across all levels of the hierarchy. Furthermore, we hypothesize that chronic drug use may exacerbate discounting, creating a feedback loop that amplifies drug-seeking tendencies. Individuals

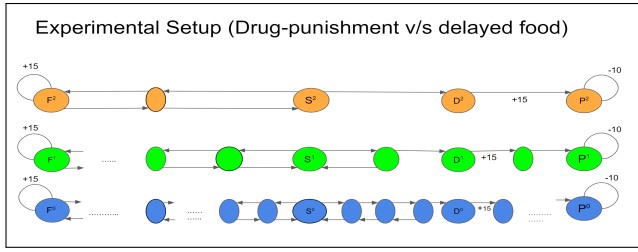

(a)

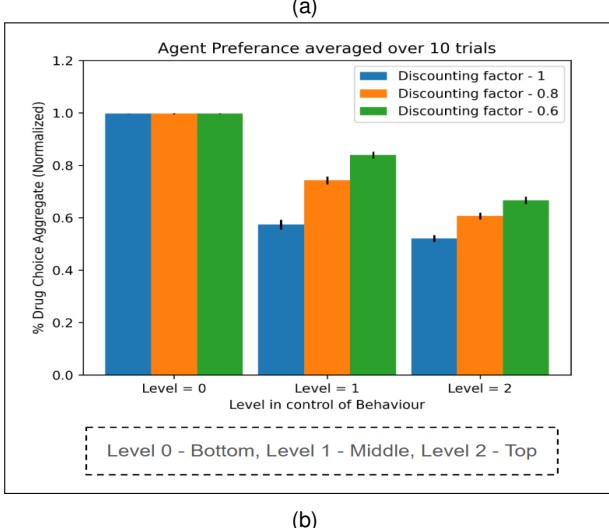

(b)

Figure 5: (a) The two-choice task environment where the agent chooses between a Food reward of $+15$ vs. a Drug reward, which is necessarily followed by a punishment of $-10$. Thus, the net reward for the drug action is $15 - 10 = 5$. (b) The bar chart illustrates the comparison of Results show a trend similar to those in Figure 4a, where drug-seeking increases when lower levels of the hierarchy control behavior. When discounting is increased, drug-seeking significantly increases at all hierarchical levels.

with inherently high baseline discounting, shaped by biological and genetic factors, may thus face a greater risk of addiction. It is also interesting to compare discounting levels in abstinent individuals with dependent users and non-users. Empirical evidence on delay discounting in abstinent individuals has shown mixed patterns across different substances. In opioid dependence, long-term abstinent individuals display similar levels of discounting to those of actively using patients—both elevated compared to non-users (Robles et al., 2011). In contrast, abstinent individuals with a history of alcohol or nicotine use tend to show reduced discounting, indicating partial recovery of future-oriented decision-making (Bickel et al., 1999; Petry, 2001). A hierarchical reinforcement learning framework offers a potential explanation: impulsivity is influenced by the level of the hierarchy currently governing behavior. We speculate that in abstinent opioid users, behavioral control may remain biased toward levels closer to the lower layers of the hierarchy, reflecting higher impulsivity, whereas alcohol and nicotine abstinence may allow re-engagement of higher-level,

more deliberative control. Future studies would be required to test this idea, for example neuroimaging approaches may be able to examine which regions of the decision hierarchy are active during inter-temporal choice across substance types and stages of abstinence.

One line of research suggests that the neuromodulator serotonin plays a crucial role in regulating temporal discounting and impulsivity in individuals (Schweighofer et al., 2008; Miyazaki et al., 2012). Specific serotonergic adaptations associated with controlled drug use can increase vulnerability to compulsive drug-seeking behaviors (Müller & Homberg, 2015). Future extensions of formal addiction models can integrate serotonin levels by modeling their influence on temporal discounting and impulsivity, further refining our understanding of their role in addiction.

In line with previous works using the biological HRL framework to explain drug seeking (Keramati & Gutkin, 2013; Mahajan et al., 2023), our model highlights how drug-induced misvaluations, particularly at lower hierarchical levels, can drive drug-seeking behavior. Control allocated to these lower levels, biased by drug-related prediction errors, promotes compulsive habits. Our model suggests, supported by prior simulation results from Mahajan et al. (2023), that individuals with SUDs may disproportionately rely on lower-level mechanisms, compared to non-addicted individuals who retain greater engagement of higher-level cognitive processes. Empirical studies comparing basal ganglia–cortical interactions in addicted versus non-addicted populations would provide valuable validation for this theoretical prediction.

Relapse can be viewed as the reactivation of dormant, low-level maladaptive habits that were previously suppressed by higher-level cognitive control during therapy or abstinence. However, dysfunction at the top levels may reduce—but not eliminate—the likelihood of addictive behaviors. Stress or drug re-exposure may further impair cognitive control, enabling habit-driven drug seeking to resurface. Future work should investigate how control is allocated across hierarchical levels. Prior studies suggest a cost-benefit arbitration mechanism may govern this process (Kool et al., 2017; Mahajan et al., 2023).

However, our model is not intended to provide a complete account of drug addiction. Many unexplained aspects of addiction involve other brain systems known to be affected by drugs of abuse (Koob, 2015). Further, our model does not capture the contributions to addiction from a goal-directed system (Hogarth, 2020). In our model, drug-induced biases accumulate over DA spirals within a model-free value-estimation framework (habit learning). One potential method of including a model-based decision system within such a biological HRL framework is by having it interact with the topmost level in a Dyna-like fashion (Sutton, 1991; Mahajan et al., 2023). Incorporating these systems into a formal computational framework remains a promising direction for future research. Another limitation of our work is that the "undiscounting" step across hierarchical levels may reduce the biological plausibil-

ity of our model; empirical studies are required to validate how inter-level communication occurs biologically. Additionally, a direct comparison between the model's discounting behavior and empirical data from individuals with SUD remains an important direction for future research.

In summary, this work extends previous HRL models by incorporating temporal discounting, offering a more comprehensive framework for understanding addiction-related behaviors. Our findings emphasize the critical role of hierarchical misvaluations and discounting in driving drug-seeking tendencies, with lower levels of the decision-making hierarchy playing a key role in compulsive habits. The model aligns with behavioral evidence linking over-discounting rates to addiction severity and highlights relapse as a reactivation of latent maladaptive habits under diminished cognitive control.

## Methods

### Computational Theory of Biological HRL

In our model, the agent updates its value estimates $Q(s_t, a_t)$ using temporal-difference (TD) learning (Sutton & Barto, 1998), computing reward prediction errors upon receiving reward $r_t$ for action $a_t$ in state $s_t$:

$$\delta_t = r_t + \gamma V(s_{t+1}) - Q(s_t, a_t) \tag{6}$$

where $V(s_{t+1})$ is the next state's value and $\gamma$ is the discount factor.

Hierarchical decision-making organizes actions with temporal and state abstractions, decomposing high-level goals into lower-level sub-goals and primitive actions (Figure 2a). Updates occur upon completion of primitive actions, ensuring convergence at the goal states. Higher-level abstractions facilitate faster learning by refining teaching signals for lower levels, accelerating convergence:

$$\delta_t^n = r_t^n + V^{n+1}(s_{t+1}^{n+1}) - Q^n(s_t^n, a_t^n) \tag{7}$$

The corresponding level is updated using:

$$Q^n(s_t^n, a_t^n) \leftarrow Q^n(s_t^n, a_t^n) + \alpha \delta_t^n \tag{8}$$

where $\alpha$ is the learning rate. This hierarchical communication aligns with the biological structure of dopaminergic spirals (Haruno & Kawato, 2006; Keramati & Gutkin, 2013; Mahajan et al., 2023).

Keramati & Gutkin (2013) modified this approach to incorporate drug-induced dopamine elevation (Di Chiara & Imperato, 1988; Redish, 2004; Dezfouli et al., 2009; Piray et al., 2010; Dayan, 2009), introducing a positive bias $d = +D$ in the prediction error:

$$\delta_t^n = r_t^n + Q^{n+1}(s_t^{n+1}, a_t^{n+1}) - r_t^{n+1} - Q^n(s_t^n, a_t^n) + d \tag{9}$$

Here, $d = 0$ for natural rewards and $d = +D$ for drug rewards. This bias leads to hierarchical misvaluation, overvaluing drug-seeking actions at lower levels. Mahajan et al. (2023) further extended this model to translate these misvaluations into behavioral choices.

### Arbitration scheme and action selection

In our experiments, we use hard allocation, where control is clamped to a single level in the hierarchy to examine behavior under full control of option selection. The controlling level selects actions via Boltzmann exploration based on Q-values and directs lower levels to execute the chosen option as a sequence of primitive actions. Reward feedback updates values across all levels. While Mahajan et al. (2023) propose a cost-benefit arbitration-based soft allocation scheme, we do not implement it here for simplicity but it can be used for future work.

### Algorithmic implementation and Simulation details

The original algorithm employed temporal and state abstractions using (1) a stacked MDP framework, (2) an option-level eligibility table, and (3) an abstract state mapping table Mahajan et al. (2023). We extend this by introducing a **stacked discounting map** that encodes each state's distance to terminal states within each hierarchical level, which is essential for propagating undiscounted state and action values downward. Value updates occur at all hierarchy levels associated with the chosen option. Rewards are assigned only once per episode—either upon final action completion or upon entering a drug-reward state followed by punishment and episode termination—consistent with the assumption that abstract actions yield rewards only upon completion. Thus, the discount factors $\zeta(n)$ are selected to ensure that natural reward values remain consistent across hierarchical levels Keramati & Gutkin (2013).

Simulations are performed in a multi-step two-choice task, averaging agent behavior over 10 random-seed trials. Each trial includes 1500 episodes, with a maximum of 400 primitive steps per episode, typically ending earlier upon reaching a terminal state. All values are initialized to zero; the learning rate is set to 0.1, and Boltzmann exploration uses a temperature of 10. Following Mahajan et al. (2023), agent performance is evaluated using cumulative termination ratios across trials to reflect reward preferences. The choice of $d = 3$ in our simulations follows the methodology of Mahajan et al. (2023), who used $d = 2$ for a drug reward of +10. Since our setup uses a drug reward of +15, we scaled $d$ proportionally, assuming a linear relationship between the bias term and drug reward magnitude.

### Explanation of steps-to-reward assumption

In our model of discounting in biological HRL, $\nu(s_t^n, a_t^n)$ represents the number of steps state $s_t^n$ is far from the goal state of action $a_t^n$ (Detailed in Figure 6). The Q-values for each state-action pair $Q(s_t, a_t)$ and the state-function values $V(s_t)$ for each state at every level converge to their discounted values according to the discount factor $\zeta(n)$ of the corresponding level. However, through equation 7, every layer passes down undiscounted values i.e values if $\zeta(n) = 1$ to the layer below it and so on. This form of information-sharing requires the user to be aware of how far the reward is in the future during the

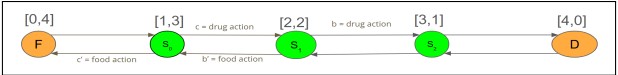

Figure 6: Illustration of the Steps-to-Reward ($\nu$) concept. In state $S_0$, taking action $c'$ (food action) leads directly to the food reward $F$ in 1 step, so $\nu(S_0, c') = 1$. However, taking action $b$ (drug action) requires 3 steps to reach the drug reward $D$: $S_0 \to S_1 \to S_2 \to D$. Therefore, $\nu(S_0, b) = 3$.

calculation of the RPE. This may be biologically plausible, as recent studies have suggested that mid-brain dopamine neurons also reflect information about the distribution of future rewards on cue presentation (Sousa et al., 2023).

## Acknowledgements

Boris S. Gutkin was funded by Agence Nationale pour la Recherche (ANR-17-EURE-0017, ANR-10IDEX-0001-02, CogFinAlgent), ENS, CNRS and INSERM.

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
