# OpenReview forum: "Discounting and Drug Seeking in Biological Hierarchical Reinforcement Learning"
_ccneuro.org/CCN/2025/Proceedings — CCN 2025 Proceedings asProceedingsPoster_

### Official Review · Reviewer_q2oG · 2025-03-14
**Interesting paper for the domains of hierarchical RL models and substance use**

**Soundness:** 2
**Clarity:** 3

**Comments:**

The paper extends an existing hierarchical reinforcement learning (HRL) model of substance use disorder (SUD) by implementing temporal discounting.  By implementing this, the authors could capture the pattern of high discounting amplifying drug-seeking, which supports a link that has been previously suggested in the literature.

The small changes in the manuscript during the revision made it easier for the naive reader to appreciate the impact of this contribution. I think it could now be of interest to many people working on hierarchical reinforcement learning (HRL) or on other models of substance use disorder.

----
Old review:

The paper extends an existing hierarchical reinforcement learning (HRL) model of substance use disorder (SUD) by implementing temporal discounting. Their main findings suggest that high discounting amplifies drug-seeking, which supports a link that has been previously suggested in the literature.

I want to start this review by saying that I do not have a strong background in reinforcement learning or substance use disorders. Other reviewers are likely more qualified to evaluate the impact of this contribution (the extension of the HRL framework) has for modeling SUB. I think the topic could be of interest to the broader CCN community, but the current state of the paper made it difficult for me to judge the novelty and impact of the contribution. Note that this is the reason why I gave the manuscript a low clarity rating: It is unclear in its aims and contribution, not in the way it reports the methods.

It seems like much of the paper is spent motivating the underlying model and its neurobiological plausibility (which was already published in Mahajan et al. 2023; as cited by the authors) rather than the changes to the model and analysis conducted, which is misleading the naïve reader regarding the aim of the manuscript.  The paper could be substantially improved by focusing only the background needed to understand the basic structure of the model.

Same is true for the discussion, where many of the conclusions drawn for substance use disorder that can be drawn (and *were* already drawn in the original Mahajan et al. paper) without the implementation of temporal discounting. From my understanding, the main contribution of the current work is the link between an increase in temporal discounting in the model and the strength in substance use in the model. While I think this could be an interesting conclusion, it must be discussed in more detail:

(1) The manuscript is missing an in-depth discussion of the temporal discounting model findings and how it relates to SUD: has this link been shown in modeling work before? How did the analysis conducted *specifically* in this manuscript advance our understanding of SUD? Does this model offer novel predictions?

(2) These novel contributions are currently buried between other (previously published) conclusions that aren’t specific to the temporal discounting implemented here. Therefore, the manuscript can be improved by being more transparent about which conclusions are novel and which that have been previously drawn from the same (similar) model. The latter can be shortened substantially if needed for context or even removed at times (especially from the abstract).

To better understand the context and the novelty of this contribution, I read over the Mahajan et al. (2023) paper that the authors extend in this paper. I was a confused to find that large passages of this manuscript are an oftentimes almost direct quotes from the previous paper. This especially concerns the passages on the neurobiological links of the model that do not seem central to the paper (at least not to this extent, see my point above) and some of the conclusions drawn in the discussion. This needs to be addressed together with a reevaluation if these sections are even necessary.

**Expertise:**

2

**Interest:**

2

---

> ### Author Rebuttal · Authors · 2025-04-13
>
> **We would like to thank the reviewer for their time and consideration of our work. We are sorry that we could not get across how our work on HRL and addiction is broadly interesting to the CCN community. Hopefully, the revisions will have improved the clarity and the aim of the manuscript.  All the edits made to the manuscript are in blue color**.
>
> **We would like to address certain points -**
>
> **Aim of the manuscript and addressing the repetitions from the previous works** – As stated in **L [153–161]**, the primary objective of this work is to introduce a novel method for incorporating temporal discounting into a biologically grounded Hierarchical Reinforcement Learning (HRL) framework. To motivate the significance and complexity of this challenge—and to establish the biological plausibility of HRL in the context of addiction—we included relevant background on prior models, particularly those by Keramati et al. and Mahajan et al. While this led to some repetition, these components were never presented as novel contributions. Rather, they were deliberately included to ensure the manuscript is self-contained and comprehensible to readers who may not be familiar with the foundational literature in this area. Furthermore, the previously established conclusions referenced in the discussion were included to clarify the specific behavioral features of addiction that HRL is capable of capturing. In response to the reviewer’s concern, we have now shortened the relevant background text, removed redundancies where possible, and enhanced the discussion to more explicitly connect our model’s findings with experimental observations. (**Shortened sections - L [163 - 196] and L[541 - 569]**)
>
> **Model Predictions** – With respect to drug rewards, our model predicts that higher discounting rates are associated with increased drug-seeking behavior across all hierarchical levels—a pattern that aligns with empirical findings in SUD populations [1]. While this correlation is known, our model offers a novel computational account of how it arises within a hierarchical decision-making framework. We hope that the revised manuscript clearly highlights these specific contributions and their relevance to addiction research.
>
> References -
> 1. Amlung, Michael, et al. "Steep delay discounting and addictive behavior: A meta‐analysis of continuous associations." Addiction 112.1 (2017): 51-62.

---

### Official Review · Reviewer_xsfY · 2025-03-24
**The authors investigate addiction to drug use by framing it as a reward discounting problem. They adapt the hierarchical RK model for decision-making to ensure reward value consistency across abstract levels. Using a simplified two-action task to simulate food-seeking and drug-seeking behaviors, the authors demonstrate that decreasing the discount factor increases drug-seeking behavior across all hierarchical levels. This approach provides a computational perspective on addiction, linking decision-making processes to maladaptive behaviors.**

**Soundness:** 2
**Clarity:** 2

**Comments:**

Interest:
This work is relevant to the computational cognitive neuroscience (CCN) community, particularly for researchers studying reinforcement learning and decision-making processes in addiction. The use of hierarchical models to explore addiction is an innovative approach, though the authors should expand on its relevance. For instance, the manuscript would benefit from discussing, instead of a hierarchy based temporal difference (TD) error,  how a single TD error could propagate across hierarchical state representations, as proposed in recent literature (Kumar et al. 2024 bioRxiv 2024.12.12.627755). Additionally, while the problem is of broad interest, the study is limited to exploring only two hyperparameters (gamma discount factor and abstraction level). Including a third hyperparameter and demonstrating that the model recapitulates at least one behavioral or experimental result from human addiction studies would significantly enhance its relevance and impact.

Soundness: The evidence adequately supports the claims, and the methods are sufficient for the study's goals. However, several questions remain unanswered. The authors use D=3 for most simulations but do not justify this choice or explore how varying D might affect the conclusions. Furthermore, the necessity of using of a hierarchical model is not fully discussed. A comparison of hierarchical processing in addiction versus non-addiction scenarios, particularly in terms of basal ganglia-cerebrum interactions, would provide deeper insights. For example, do individuals with addiction use more lower-level processing (level 0) compared to non-addicted individuals who might use higher-level processing (level 2)? Addressing these points would strengthen the manuscript's theoretical foundation.

Clarity:
The manuscript is generally well-written, but some aspects require clarification. For instance, the inclusion of the additional d value in Eq. 1 to bias drug use is not well-justified. If the drug inherently provides a reward of +15, it is unclear why an additional bias term is necessary for value computation. Additionally, in Fig. 4, the distribution of agents (each bar graph has error bar) exhibiting drug-seeking behavior is not explained. Were the agents initialized with different Q-value tables, or does the distribution arise from stochasticity in the learning or action selection process? Providing these details would improve reproducibility and ensure readers can fully understand the model's implementation and results. Overall, the contributions are communicated effectively, but addressing these points would enhance clarity and methodological rigor.

**Expertise:**

2

**Interest:**

3

---

> ### Author Rebuttal · Authors · 2025-04-13
>
> **We thank the reviewer for their time and consideration of our work.  All the edits made to the manuscript are in blue color**
>
> **Relevance of HRL models** – Our work builds upon the hypothesis proposed by Redish[1] and later extended by Keramati & Gutkin [4]. Neurobiologically, it draws on models of TD-error propagation through cortico-striatal-cortical loops in the basal ganglia [4-6], providing a framework distinct from other hierarchical approaches such as those involving place field reorganization—where a single TD error may propagate across abstract state representations [7]. While we acknowledge alternative models of addiction, our manuscript focuses on developing a model rooted in biologically grounded HRL that explains how addiction-related misvaluations can emerge from hierarchical decision-making processes. We revised **L [87–95]** to better emphasize the relevance of this modeling approach
>
> **Hyperparameter exploration** -This paper primarily introduces discounting into the HRL framework to study its impact on addictive behaviors; thus, we focused on the two most relevant hyperparameters. Investigating additional hyperparameters is beyond our current scope but remains an important direction for future work
>
> **Hierarchical processing in addiction vs non-addiction scenarios** -
> It is implied that in people with SUDs, their behavior is mainly controlled by the lower-levels. We have addressed this in the discussion (**L [492-499]**)
>
> **Incorporation of additional d value and d=3 for simulations** -
> The addition of the +d bias term to the reward prediction error equation has been in all the model-free RL approaches to model addiction [1-5]. It is assumed to model the transient increase in dopamine through neuropharmacological mechanisms caused by cocaine and other addictive drugs
> Using d = 3 - We added this in experimental details (**L [606 - 611]**)
>
> **Explanation of Fig 4** -
> Each agent had been initialised with the same Q-table of all 0’s (**L 602**), We have added an explanation for the stochasticity observed in the caption of Fig 4
>
> References -
> 1. Redish, Science 306.5703 (2004): 1944-1947
> 2. Dezfouli et al. Neural computation 21.10 (2009): 2869-2893
> 3. Piray, Payam, et al. Neural computation 22.9 (2010): 2334-2368
> 4. Keramati, Gutkin. PloS one 8.4 (2013): e61489
> 5. Mahajan, et al. Addiction Neuroscience 8 (2023): 100115
> 6. Haruno, Masahiko, and Mitsuo Kawato. Neural networks 19.8 (2006): 1242-1254
> 7. Kumar, et al. bioRxiv (2024): 2024-12

---

### Official Review · Reviewer_LRXA · 2025-03-26
**a sound and clear description of a hierarchical RL model of substance use.**

**Soundness:** 2
**Clarity:** 3

**Comments:**

General summary:
This is a very clearly written paper describing a hierarchical reinforcement learning model of the neural basis of substance abuse disorder. The paper makes several updates to the SOTA models to better discount rewards across the reward structure. The background and the model are both clearly justified and described, though the clarity is mildly impugned by some discipline specific terminology (i.e. successors, spiraling connections) that may limit this paper's utility for a broader audience. I would have loved a little more discussion of the implications for SUD in humans. Overall, this is a sound and clear update to a hierarchical RL model of drug seeking behavior.

A few specific comments:
- The "undiscounting" step across hierarchical levels may reduce the biological plausibility of the model.
- The paper shows that the model has reduced Q-values for delayed rewards, indicating that it is carrying out normative delayed discounting, however, the papers results would be much stronger and more general to compare qualitative differences in delayed discounting between the model and individuals with SUD.
- There is some interchange between descriptions of impulsivity and compulsivity that could be clarified or disambiguated

**Expertise:**

2

**Interest:**

3

---

> ### Author Rebuttal · Authors · 2025-04-13
>
> **We thank the reviewer for their time and consideration of our work. All the edits made to the manuscript are in blue color.**
>
> **We would like to address certain points**-
>
> **Implications for SUD in Humans and Comparison with Human Data**: The main contribution of this paper was introducing a novel method of incorporating discounting in the HRL framework and having a biological grounding for the framework. The model proposes that the overall discounting in the case of natural rewards is normative but not in the case of drug rewards because of the +d factor in the reward prediction error. We agree that comparing delayed discounting between the model and individuals with SUD would enhance the work. While this is beyond the current scope, we have noted this as a direction for future research in the discussion (**L [526-529]**).
>
> **Biological plausibility of the “undiscounting” step**: We addressed this in the discussion as a possible limitation (**L [522-526]**). Thank you for pointing it out.
>
> **Impulsivity and Compulsivity**: We have now included the definitions of compulsivity and impulsivity (**L [136 - 139],  [146-148]**).

---

> > ### Comment · Reviewer_LRXA · 2025-04-18
> >
> > The authors have adequately addressed my few minor comments.

---

### Meta-Review · Area_Chair_4KDL · 2025-05-06

**Ccn Recommendation:** Accept as Proceedings

**Metareview:**

There was a general consensus among reviewers that the work is methodologically sound and of broad interdisciplinary interest. Most reviewers found the paper to be clearly written, with several rating the clarity as exceptional. The major comments focused on clarifying specific methodological details and improving the exposition of how the modeling approach extends prior work. The authors responded constructively, providing revisions that addressed both the technical and clarity-related concerns. Taken together,  the overall assessment of the reviewers was positive and the reviewers’ concerns were satisfactorily resolved during the discussion period, and I am happy to recommend acceptance.

**Summary:**

All reviewers acknowledged the topic’s broad interdisciplinary appeal, and found the soundness of the work to be adequate. Most reviewers rated the clarity as exceptional, while one considered it adequate. Two reviewers raised major concerns about the initial version of the paper. One recognized the value of applying hierarchical models to the study of addiction but emphasized the need to justify specific modeling choices and simulation parameters, and requested a comparison between hierarchical processing in addiction versus non-addiction scenarios. In my view, the authors addressed these concerns satisfactorily in their revisions. Another reviewer initially found it difficult to assess the novelty of the contribution due to a lack of clarity in the original manuscript; this issue was effectively resolved in the revised version.

**Expertise:**

3